# DFMVC: Deep Fair Multi-view Clustering

## ABSTRACT

Fair multi-view clustering aims to achieve both satisfactory clustering performance and non-discriminatory outcomes with respect to sensitive attributes. Existing fair multi-view clustering methods impose a constraint that requires the distribution of sensitive attributes to be uniform within each cluster. However, this constraint can lead to mis-allocation of samples with sensitive attributes. To solve this problem, we propose a novel **D**eep **F**air **M**ulti-**V**iew **C**lustering (DFMVC) method that learns a consistent and discriminative representation instructed by a fairness constraint constructed from the distribution of clusters. Specifically, we incorporate contrastive constraints on semantic features from different views to obtain consistent and discriminative representations for each view. Additionally, we align the distribution of sensitive attributes with the target cluster distribution to achieve optimal fairness in clustering results. Experimental results on four datasets with sensitive attributes demonstrate that our method improves both the fairness and performance of clustering compared to state-of-the-art multi-view clustering methods.

## CCS CONCEPTS

• **Computing methodologies** → **Cluster analysis**; • **Security and privacy** → **Privacy protections**.

## KEYWORDS

Multi-view clustering, Fair clustering

## 1 INTRODUCTION

With the rapid development of digital technologies, data often consists of diverse features or originates from various views. For example, an image can be described by diverse feature sets, including Local Binary Patterns (LBP), Histogram of Oriented Gradients (HoG) and Scale-Invariant Feature Transform (SIFT), and each feature set can represent a view of an image. Multi-view clustering (MVC), as an unsupervised learning task, has gained considerable attention in various machine learning applications, including data mining [50], scene recognition [34] and information retrieval [15]. Compared to single-view clustering, because of the complementary nature of information between different views [5, 34], MVC significantly enhances clustering performance by integrating information from multiple views. The goal of MVC is to divide data samples into several disjoint groups by using feature information between different views.

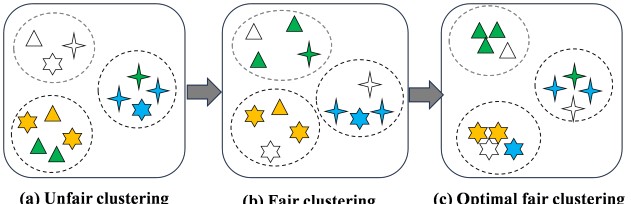

(a) Unfair clustering     (b) Fair clustering     (c) Optimal fair clustering

**Figure 1: An example illustration of the motivation. Different shapes represent different classes. Colored shapes represent samples with sensitive attributes, and uncolored shapes represent samples without sensitive attributes. Most fair multi-view clustering focuses on the uniform distribution of sensitive distribution of sensitive attributes in each cluster while ignoring the the true cluster distribution structure of samples with sensitive attributes.**

Existing MVC algorithms can be roughly categorized into traditional MVC methods and deep MVC methods. Traditional MVC methods contain three categories: non-negative matrix factorization (NMF)-based methods [30], graph-based methods [24, 26], and subspace-based methods [53, 56]. NMF-based MVC employs matrix factorization to learn a shared representation for multi-view data. Graph-based MVC methods construct a unified graph structure using multi-view data. Subspace-based MVC methods aim to learn a consistent subspace representation. However, these traditional MVC methods primarily rely on linear transformations, limiting their potential to improve clustering performance.

Deep MVC algorithms have been proposed to address the aforementioned limitation, leveraging the powerful nonlinear feature extraction capabilities of deep neural networks [11, 36, 48, 49]. These methods employ view-specific encoder networks to transform each view and derive a consensus representation. For example, MvDSCN [57] learned a multi-view self-representation matrix in an end-to-end manner. Inspired by contrastive learning techniques, many multi-view clustering methods integrate this technique into the model. Most existing contrastive learning-based methods attempt to maximize the mutual information contained among the assignment distributions of multiple views. For instance, COMPLETER [29] learned an informative and consistent representation of multi-view data by contrastive learning; DCP [28] conducted consistency learning and data recovery through dual contrastive prediction; MFLVC [49] introduces multi-level feature learning with a contrastive strategy for multi-view clustering, e.g., low-level features, high-level features and semantic features. Although these methods achieve remarkable clustering performance, it should be noted that multi-view data often contains sensitive attributes, which can influence the clustering outcome. For instance, if the sensitive attribute is gender, one cluster may predominantly consist of males while another cluster predominantly consists of females, resulting

in gender bias. In Figure 1(a), we can see that it will lead to unfair clustering.

To alleviate the unfairness issue, numerous fair clustering methods have been proposed [10, 20, 22]. For instance, fair clustering was first proposed and divided the original data into several chunks with fairness constraints in [10]. In Figure 1(a) and (b), fair clustering can ensure the uniform distribution of samples containing sensitive attributes in each cluster. SpFC [20] imposed linear fairness constraints on spectral clustering. DFC [22] exploited adversarial training to make the clustering results independent of sensitive attributes. FCMI [52] implemented fair clustering by mutual information theory, maximizing mutual information between non-group information and cluster information and minimizing mutual information between group information and cluster information. Fair-MVC [55] explored the fairness issue on multi-view scenarios by enforcing the distribution of sensitive attributes within each cluster to be uniform. However, it excessively emphasizes fairness constraints but neglects the true cluster distribution structure of samples with sensitive attributes, leading to incorrect clustering assignments for them. As shown in Figure. 1(b) and (c), most fair clustering methods force the samples containing sensitive attributes in each cluster to be evenly distributed, while ignoring the clustering structure of the samples themselves.

In mitigating the challenge, in this paper, we propose a deep fair multi-view clustering method named DFMVC, whose framework is shown in Figure. 1. We initially leverage deep autoencoder to extract view-specific features from different views. Then, to further exploit the discriminative and complementary information of the multi-view data, we transform the features from different views into semantic spaces and leverage contrastive learning to obtain consistent semantic representations. Finally, in order to achieve optimal fairness in clustering results while ensuring that sensitive attributes do not impact the clustering performance, we design a target cluster distribution-guided fairness learning module. In this module, we learn a target cluster distribution by fusing view-specific representations. Additionally, we utilize a constraint based on KL-divergence to align the distribution of sensitive attributes with the target cluster distribution to achieve optimal fairness in clustering results. We summarize the main contributions of this paper as follows:

- We leverage target cluster distribution to guide fair clustering and propose a novel deep fair MVC algorithm termed DFMVC. It makes clustering results independent of sensitive features, alleviating unfair issues in MVC.
- By contrasting the semantic representation across multiple views, our method explores discriminative and consistency features among views.
- Extensive experiments on four datasets have demonstrated the superiority of the proposed DFMVC method. Furthermore, the effectiveness of the proposed modules is verified by ablation studies.

## 2 RELATED WORK

### 2.1 Multi-view Clustering

In recent years, MVC [8, 38, 41, 42, 49] has gained considerable attention, with a growing body of research addressing this area.

The current multi-view clustering method can be broadly classified into two main categories: traditional MVC methods and deep MVC methods. Traditional MVC methods can be divided into three main categories: (1) Non-negative matrix factorization-based methods [54]. It usually utilizes matrix factorization to decompose the integrated feature matrix, aiming to learn a shared representation. [45] introduced gradual factorization of multi-view data matrices into representational subspaces lay-by-layer and generated one clustering in each layer. (2) Subspace-based methods, aiming to learn consistent subspace representation across multiple views. For instance, [41] proposed a deep structured multi-pathway network for multi-view subspace clustering task. (3) Graph-based methods [32, 35] utilize multi-view data to construct graph structure. However, these traditional methods tend to learn shallow representations of multi-view data, thereby constraining the discriminative capability of the obtained representations.

Benefiting from deep learning, deep neural networks have the power ability to extract feature representations. In recent years, many deep multi-view clustering methods have been proposed [18, 27, 29, 49]. First, one of the most representative deep MVC methods is based on deep autoencoders. These methods aim to learn consensus representation by minimizing the reconstruction loss by instances of multiple views. DMSC-UDL [40] employed deep autoencoder to obtain representations of different views, imposing different constraints to obtain self-expression layers. Second, contrastive learning-based MVC methods. By maximizing the distance between positive pairs and minimizing the distance between negative pairs, MVC can learn more discriminative clustering information. AGCL [44] explored intra-cluster consistency and inter-view consistency through within-view graph contrastive learning and cross-view graph consistency learning. Last, adversarial network-based MVC methods employ adversarial training to further capture the data distribution. DAMC [25] utilized adversarial training to capture the data distribution and disentangle the latent space. These methods achieve better performance.

### 2.2 Fair Clustering

Recently, there has been significant interest in the machine learning community regarding the topic of fairness in clustering. To mitigate or even eliminate the impact of sensitive attributes, many efforts have been devoted to fair clustering[1, 4, 7, 9]. Based on how fairness is incorporated, the existing works can be roughly divided into three categories: (1) Pre-processing method. It transforms the original data to satisfy fairness constraints and uses classic clustering algorithms to achieve fair clustering. For example, [10] first divided the original data into several chunks with fairness constraints (i.e., fairlets), and then used the classic clustering algorithm on these fairlets to obtain fair clusters. However, constructing the fairlets requires at least quadratic running time, which poses a significant challenge in practical applications. Therefore, ScFC [2] proposed a tree metric to construct fairlets in near-linear time. (2) In-processing methods. Different from pre-processing methods, it adds fairness as a constraint to the objective function for joint optimization. For instance, SpFC [20] treated fairness as a constraint and integrated it into spectral clustering. (3) Post-processing methods [3]. In contrast to pre-processing methods, post-processing methods techniques

**Table 1: Basic notations used in the whole paper.**

| Notation | Meaning |
|---|---|
| $\mathbf{X}^v$ | The input data matrix of $v$-th view |
| $\tilde{\mathbf{X}}$ | The reconstruction data matrix of $v$-th view |
| $\mathbf{Z}^v$ | The embedded feature of $v$-th view |
| $\mathbf{H}^v$ | The embedded feature of $v$-th view |
| $\mathbf{S}$ | The sensitive feature |
| $\mathbf{Z}^G$ | The global feature of all views |
| $\mathbf{V_n}$ | The view number of the multi-view dataset |
| $\mathbf{N}$ | The number of samples |
| $\mathbf{d_v}$ | The dimension of input data matrix |
| $\mathbf{d_z}$ | The dimension of embedded feature |
| $\mathbf{K}$ | The number of cluster |
| $\mathbf{q}_{ik}$ | Soft clustering assignment |
| $\mathbf{p}_{ik}$ | Target assignment |

focus on converting the existing clustering result into a fair result through the resolution of a linear programming problem.

Inspired by the achievement of deep clustering [12–14, 17], deep fair clustering has attracted widespread attention. For instance, Towards [39] proposed a method to learn a fair embedding (i.e., fairoid) by enforcing cluster centroids to be equi-distant from each fairoid. DFC [22] explored fair clustering by adversarial training. FCMI [52] researched deep fair clustering by mutual information theory. However, these works mainly focus on a single view. Fair-MVC [55] delved into fair clustering on multi-view data, but this method limits the clustering performance by forcing the fraction of different groups in each cluster to be approximately identical to the entire dataset.

## 3 METHODOLOGY

In this section, we introduce a novel Deep Fair Multi-view Clustering. The comprehensive DFMVC framework is illustrated in Figure 1. DFMVC mainly consists of three key modules: Multi-view Data Reconstruction Module, Semantic Contrastive Learning Module, and Cluster Distribution-guide Fair Learning Module. Specific descriptions of these modules will be provided in the subsequent sections.

### 3.1 Notations

Given a set of multi-view data $\mathcal{X} = \{X^1, X^2, ..., X^{V_n}, S\}$, where $X^v \in \mathbb{R}^{N \times d_v}(v = 1, 2, ..., V_n)$ is the input data matrices for the $v$-th view. $S \in \mathbb{R}^{N \times d_s}$ is the sensitive features (e.g., gender, race, etc.). $V_n$ is the number of views, $N$ is the number of samples, $d_v$ is the dimension of $X^v$, and $d_s$ is the dimension of sensitive features. Assume that $K$ is the number of clusters. The samples with the same semantic labels can be grouped into the same cluster. Therefore, $N$ samples can be categorized into $K$ different clusters. A summary of essential notations is provided in Table 1.

### 3.2 Multi-view Data Reconstruction Module

Due to original multi-view data usually containing redundancy and random noise, we need to initially learn representative feature representations from the original data. In particular, the autoencoder [16, 33] stands out as a commonly employed unsupervised model capable of transforming original data features into a low-dimensional feature space. Specifically, for the $v$-th view, let $f_{\theta^v}^v(\cdot)$ $(1 \leq v \leq V_n)$ denote the encoder nonlinear function. In the encoder, the network can learn the low-dimensional features as follows:

$$\mathbf{z}_i^v = f_{\theta^v}^v(\mathbf{x}_i^v) \tag{1}$$

where $\mathbf{z}_i^v \in \mathbb{R}^{d_z}$ is the embedded feature representation in $d_z$ dimensional feature space of $i$-th sample from the $v$-th view $\mathbf{x}_i^v$.

Then, the decoder reconstructs the sample by the feature representation $z_i^v$. Let $g_{\phi^v}^v(\cdot)$ $(1 \leq v \leq V_n)$ represent the decoder function. In the decoder part, the reconstructed sample $\tilde{\mathbf{x}}_i^v$ is obtained by decoding $\mathbf{z}_i^v$:

$$\tilde{\mathbf{x}}_i^v = g_{\phi^v}^v(\mathbf{z}_i^v) = g_{\phi^v}^v(f_{\theta^v}^v(\mathbf{x}_i^v)) \tag{2}$$

Let $\mathcal{L}_r$ be the reconstruction loss from input $\mathbf{X}^v \in \mathbb{R}^{n \times d_v}$ to output $\tilde{\mathbf{X}}^v \in \mathbb{R}^{n \times d_v}$, $n$ denote the number of samples in a batch. The reconstruction loss is formulated as:

$$\begin{aligned}\mathcal{L}_r &= \sum_{v=1}^{V_n} \mathcal{L}_r^v = \sum_{v=1}^{V_n} \left\| \mathbf{X}^v - \tilde{\mathbf{X}}^v \right\|_2^2 \\ &= \sum_{v=1}^{V_n} \sum_{i=1}^{n} \left\| \mathbf{x}_i^v - g_{\phi^v}^v(\mathbf{z}_i^v) \right\|_2^2 \end{aligned} \tag{3}$$

During the pre-training stage, we utilize the reconstruction loss function for training, aiming to initialize the parameters. Each encoder and decoder consists of four layers, and the nonlinear rectified linear unit (ReLU) function is chosen as the activation function in the deep autoencoder.

### 3.3 Semantic Contrastive Learning Module

In this subsection, we will introduce a semantic contrastive learning module to obtain consistent and discriminative features. Based on $\mathbf{Z}^v = f(\mathbf{X}^v; \theta^v)$, we can obtain embedded features. We stack two linear layers and a successive softmax function on embedded features to produce a cluster assignment, which is computed by: $f_{W^v} : \{\mathbf{Z}^v\} \rightarrow \{\mathbf{H}^v\}$ $(1 \leq v \leq V_n)$, where $W^v$ is the learnable parameters of linear layers.

Inspired by recently proposed contrastive learning techniques [6], we can leverage these methods on the embedded features to explore consistency across multiple views. By doing so, We can acquire cluster assignment probability matrices $\mathbf{H}^v \in \mathbb{R}^{N \times K}$($1 \leq v \leq V_n$) for all views, where $K$ is the number of clusters. Let $\mathbf{h}_i^v$ denote the $i$-th row in $\mathbf{H}^v$, $\mathbf{h}_{ij}^v$ represent the probability that instance $i$ belongs to cluster $j$ in the $v$-th view, and $\mathbf{h}_j^v$ represents a cluster assignment of the same semantic cluster. The semantic label of instance $i$ is determined by the highest probability within $\mathbf{h}_i^v$. The instances should share the same semantic information. we define the similarity between two cluster assignments $\mathbf{h}_j^{v_1}$ and $\mathbf{h}_j^{v_2}$ of cluster $j$ as

$$sim(\mathbf{h}_j^{v_1}, \mathbf{h}_j^{v_2}) = \left(h_j^{v_1}\right)^T h_j^{v_2} \tag{4}$$

where $\langle \cdot, \cdot \rangle$ is dot product operator, $v_1$ and $v_2$ denote two different views. In our method, the cluster probabilities of instances among different views should be similar, as these instances collectively

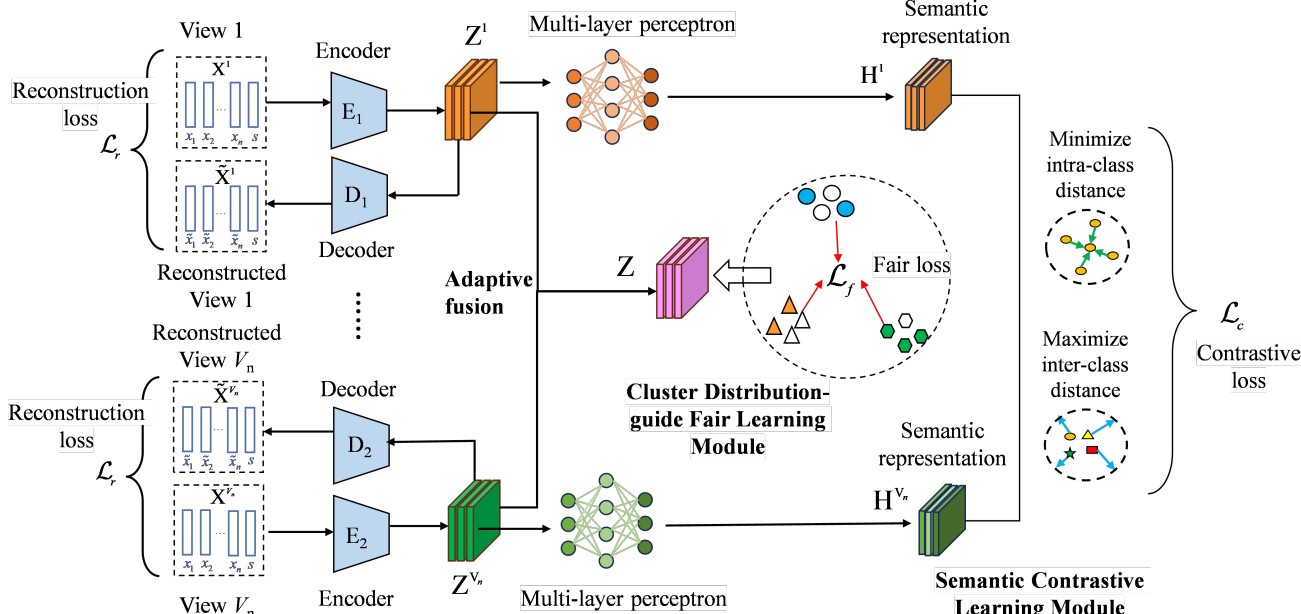

**Figure 2: Illustration of DFMVC. In our method, we design a cluster distribution-guide fair learning module and semantic multi-view contrastive learning module. The former aims to align the distribution of sensitive attributes with the target cluster distribution. The latter explores semantic consistency learning among different views and obtains discriminative and consistent representations by contrasting clustering assignments among different views. The multi-layer perceptron (MLP) consists of multiple linear layers. The view-specific autoencoder contains the encoding part and the decoding part.**

characterize the same sample. In addition, the instances in multiple views should be independent of each other, as they are utilized to characterize different samples. Therefore, for the $v$-th view, the same semantic labels $\mathbf{h}_j^v$ should have $(V_n - 1)$ positive cluster assignment pairs and $V_n(K - 1)$ negative cluster assignment pairs when considering $\mathbf{h}_j^v$ and $K$ cluster across $V_n$ views. Specifically, we consider instances of the same sample across different views as positive pairs, instances of different samples within the same view, as well as instances across different views, as negative pairs.

The similarities among positive pairs should be maximized, and those among negative pairs should be minimized. The semantic contrastive loss between $\mathbf{h}_j^{v_1}$ and $\mathbf{h}_j^{v_2}$ is defined as follows:

$$l^{(v_1 v_2)} = -\frac{1}{K} \sum_{j=1}^{K} \log \frac{e^{sim(\mathbf{h}_j^{v_1}, \mathbf{h}_j^{v_2})/\tau_L}}{T},$$

$$T = \sum_{k=1, j \neq k}^{K} e^{sim(\mathbf{h}_j^{v_1}, \mathbf{h}_k^{v_1})/\tau_L} + e^{sim(\mathbf{h}_j^{v_1}, \mathbf{h}_k^{v_2})/\tau_L} - e^{1/\tau_L} \tag{5}$$

where $\tau_L$ is a temperature parameter, $(\mathbf{h}_j^{v_1}, \mathbf{h}_j^{v_2})$ is a positive cluster assignment pair between $v_1$-th view and $v_2$-th view, $(\mathbf{h}_j^{v_1}, \mathbf{h}_k^{v_1})$ is a negative cluster assignment pair between $j$-th cluster and $k$-th cluster in the $v_1$ view, and $(\mathbf{h}_j^{v_1}, \mathbf{h}_k^{v_2})$ is also a negative cluster assignment pair between $v_1$-th view and $v_2$-th view. Therefore, semantic

contrastive loss included across multiple views could be defined as:

$$\mathcal{L}_c = \frac{1}{2} \sum_{v_1=1}^{V_n} \sum_{v_2=1, v_2 \neq v_1}^{V_n} l^{(v_1, v_2)} + \sum_{v=1}^{V_n} \sum_{j=1}^{K} r_j^v \log r_j^v \tag{6}$$

where $r_j^v = \frac{1}{N} \sum_{i=1}^{N} h_{ij}^v$. The first part of Eq. (6) aims to learn the clustering consistency for all views. It pulls pairs of cluster assignments from the same cluster closer together and pushes cluster assignments from The second part of Eq. (6) is a regularization term, which is used to prevent all samples from being assigned exclusively to a single cluster. Specifically, if $h_{ij}^v = 1$ for all $i = 1, 2, ..., N$, it means that all instances only belong to $j$-th cluster, at this time, $r_j^v$ is 1, therefore, $r_j^v \log r_j^v$ is 0. When $0 \leq h_{ij}^v \leq 1$, at this time, $r_j^v \log r_j^v \leq 0$. By this regularization term, we can make sure that each cluster has at least one sample. Hence, this loss can encourage the acquisition of clustering information that is both discriminative and consistent across multiple views.

### 3.4 Cluster Distribution-guide Fair Learning Module

In this subsection, we will introduce a cluster distribution-guide fair learning module to accomplish fair clustering. Inspired by other deep clustering [22, 46], we employ target cluster distribution $P$ to guide the learning of sample distributions with sensitive attributes. Specifically, we design a weighted adaptive fusion mechanism, aiming to fuse the embedded features from each view into a unified embedded feature. We will learn $V_n$ embedded feature for each

**Algorithm 1** The optimization of DFMVC

**Input:** The multi-view data matrices $\mathcal{X}$, the iteration number $I$, parameters $\alpha$ and $\beta$.

**Output:** The predicted labels $\mathbf{Y} = [y_1, y_2, ..., y_n]$.

1: Initialize the weight by minimizing $\mathcal{L}_r$ in Eq. (3);
2: **for** $i = 1$ to $I$ **do**
3:   Random select a minibatch of samples;
4:   Computing $\mathbf{Z}^v$ and $\mathbf{Z}^G$ by Eqs. (1) and (7), respectively;
5:   Computing $q_{ik}$ and $p_{ik}$ by Eqs. (8) and (9), respectively;
6:   Computing $\mathcal{L}_r, \mathcal{L}_c, \mathcal{L}_f$ by Eqs. (3), (6) and (10)
7:   Optimizer parameters by Eqs. (11);
8: **end for**
9: Calculate the semantic labels by Eqs. (12).

view data, i.e., $\{\mathbf{Z}^1, \mathbf{Z}^2, ..., \mathbf{Z}^{V_n}\}$, where $\mathbf{Z^v} = f(\mathbf{X^v}, \theta^v)$. We get a common representation $\mathbf{Z}$ based on all embedded features. It can be calculated as:

$$\mathbf{Z} = \frac{\sum_{v=1}^{V_n} a_v \mathbf{Z}^v}{\sum_{v=1}^{V_n} a_v} \tag{7}$$

where $a_v \in A$, $A = [a_1, a_2, ..., a_{V_n}]$, $\mathbf{Z}^v \in \mathbb{R}^{n \times d_z}$, $\mathbf{Z} \in \mathbb{R}^{n \times d_z}$, $n$ denote the number of samples in a batch. $\mathbf{Z}$ represents the global embedded feature obtained after fusing each view. For example, if we have two views, it can write that $\mathbf{Z} = \frac{a_1 \mathbf{Z}^1 + a_2 \mathbf{Z}^2}{a_1 + a_2}$. $A$ is a set of learnable parameters, and it will learn an appropriate weight for each view. If a view contains more information, the greater its corresponding weight. Then, we can define soft assignment $q$ as follows:

$$q_{ik} = \frac{(1 + ||\mathbf{z}_i - c_k||/\alpha)^{-\frac{\alpha+1}{2}}}{\sum_{k' \in [K]} (1 + ||\mathbf{z}_i - c_{k'}||/\alpha)^{-\frac{\alpha+1}{2}}} \tag{8}$$

where $c_k$ represents the cluster centroid, $\alpha$ is the degree of freedom of Student's t-distribution. In our experiment, we set $\alpha$ to 1. $q_{ik}$ can be interpreted as the probability the $i$-th sample is assigned to the $k$-th cluster.

We follow [46] to use Student t-distribution for assignment, but the difference is that performing this operation on each protected subgroup to prevent any single cluster from being dominated by samples with the same sensitive attribute. The auxiliary target distribution $P$ is calculated by:

$$p_{ik} = \frac{(q_{ik})^2 / \sum_{x^v \in X_g} q_{ik}}{\sum_{k' \in [K]} \left( (q_{k'}^2) / \sum_{x^v \in X_g} q_{k'} \right)} \tag{9}$$

where $X_g$ is the subgroup that the instance belongs to. For example, if the sensitive feature is gender, it has two subgroups (males and females). Through this process, we can obtain target cluster distribution. The cluster distribution-guide fair loss is KL divergence between soft assignment and auxiliary target distribution:

$$\mathcal{L}_f = KL(P||Q) = \sum_{s \in [S]} \sum_{x^v \in X_g} \sum_{k \in [K]} p_{ik} \log \frac{p_{ik}}{q_{ik}} \tag{10}$$

By minimizing this loss, making $q_{ik}$ gradually approach the cluster target distribution $p_{ik}$. On the one hand, it can learn more discriminative clustering information, on the other hand, it also can learn fair representations.

**Table 2: Statistics summary of four datasets.**

| Dataset | Samples | Sensitive feature | Clusters |
|---|---|---|---|
| Credit Card | 5000 | Gender | 5 |
| Bank Marketing | 5000 | Marital status | 2 |
| Law School | 3600 | Gender | 2 |
| Zafar | 10000 | Binary value | 2 |

## 3.5 Objective Function

The object function of the proposed DFMVC contains the reconstruction loss $\mathcal{L}_r$, the semantic contrastive learning loss $\mathcal{L}_c$, and the cluster distribution-guide fair learning loss $\mathcal{L}_f$. In summary, the object of DFMVC is formulated as follows:

$$\mathcal{L} = \mathcal{L}_r + \alpha \mathcal{L}_c + \beta \mathcal{L}_f, \tag{11}$$

where $\alpha, \beta$ are the trade-off parameters. The detailed learning process of our DFMVC is shown in Algorithm 1.

The proposed method aims to learn feature consistency and obtain semantic labels across multiple views. Let $h_i^v$ be the $i$-th row of $H^v$, and let $h_{ij}^v$ represent the $j$-th element of $h_i^v$. Specifically, $h_i^v$ is $K$-dimensional soft assignment probability, where $\sum_i^N h_{ij}^v = 1$. When the training process of the network is completed, the semantic label can be predicted by:

$$y_i = \arg \max_j \left( \frac{1}{V_n} \sum_{v=1}^{V_n} h_{ij}^v \right) \ (1 \leq i \leq N) \tag{12}$$

## 4 EXPERIMENT

In this section, we conduct the experiments to verify the effectiveness of the proposed DFMVC by answering the following questions:

- **RQ1**: How effective is DFMVC for deep fair clustering?
- **RQ2**: How does the proposed module influence the performance of DFMVC?
- **RQ3**: What is the clustering structure revealed by DFMVC?
- **RQ4**: How about the convergence about DFMVC?
- **RQ5**: How do the hyper-parameters impact the performance of DFMVC?

## 4.1 Datasets & Metric

**Fairness Datasets** We conduct extensive experiments to verify the effectiveness of DFMVC on four fairness datasets, including Credit Card dataset [51], Zafar dataset [51], Bank Marketing dataset [51], and Law school dataset [21]. Specifically, the Credit Card dataset describes the customers' default payments in Taiwan and this dataset consists of 5000 samples with 24 attributes. The sensitive feature is gender. The Bank Marketing dataset is associated with direct marketing campaigns of a Portuguese banking institution and it aims to see if the product (bank term deposit) would be('yes') or not ('no') subscribed. The sensitive feature in this dataset is marital status. This data set consists of 5000 instances and 17 attributes. The Law School dataset is to predict whether a candidate would pass the bar exam, and it consists of 3600 samples with 12 features. The dataset's sensitive feature is gender. Last, the Zafar dataset is a widely used synthetic dataset, where one binary value is generated

**Table 3: Results on four datasets with sensitive features. The best results are highlighted in bold, while the second-best values are underlined. (Higher balance score indicates better fairness.)**

| Methods | Credit Card | | Banking Market | | Law school | | Zafar | |
|---|---|---|---|---|---|---|---|---|
| Metrics(%) | NMI | BAL | NMI | BAL | NMI | BAL | NMI | BAL |
| K-means[31] | 20.94 ± 1.14 | 35.53 ± 0.37 | 28.67 ± 1.44 | 37.64 ± 0.66 | 20.12 ± 1.25 | 43.22 ± 1.05 | 70.32 ± 0.78 | 17.06 ± 0.76 |
| DEC[46] | 21.03 ± 2.09 | 35.96 ± 0.60 | 30.93 ± 1.15 | 37.60 ± 0.96 | 21.23 ± 1.21 | 44.15 ± 1.24 | 72.55 ± 1.92 | 16.85 ± 0.73 |
| CC[23] | 23.87 ± 1.28 | 35.74 ± 0.47 | 36.23 ± 1.01 | 37.46 ± 0.97 | 23.02 ± 0.92 | 44.24 ± 1.16 | 78.95 ± 0.68 | 17.01 ± 0.71 |
| MvDSCN[57] | 21.92 ± 1.53 | 35.82 ± 0.41 | 36.24 ± 0.55 | 37.59 ± 0.67 | 22.06 ± 1.28 | 44.82 ± 0.98 | 76.91 ± 0.42 | 17.13 ± 0.65 |
| DCP[28] | 26.73 ± 0.26 | 24.19 ± 1.05 | 39.93 ± 1.84 | 26.75 ± 2.06 | 23.85 ± 1.26 | 35.76 ± 1.12 | 81.87 ± 1.57 | 21.65 ± 1.23 |
| APADC[47] | 23.07 ± 0.45 | 26.32 ± 0.22 | 40.62 ± 0.25 | 27.79 ± 2.59 | 22.08 ± 1.24 | 36.85 ± 0.56 | 72.38 ± 0.80 | 21.21 ± 0.57 |
| MFLVC[49] | 24.02 ± 0.42 | 36.09 ± 0.24 | 37.76 ± 1.15 | 38.64 ± 1.48 | 21.84 ± 1.52 | 43.87 ± 0.46 | 90.52 ± 1.42 | 27.16 ± 0.85 |
| Fair-MVC[55] | 24.19 ± 0.51 | 37.23 ± 0.42 | 38.89 ± 0.91 | 40.75 ± 1.56 | 21.57 ± 0.92 | 44.79 ± 0.56 | 81.81 ± 0.57 | 28.32 ± 0.48 |
| DFMVC | **35.13 ± 0.62** | **39.71 ± 0.38** | **54.62 ± 1.25** | **42.16 ± 0.82** | **24.24 ± 0.32** | **45.66 ± 0.38** | **93.93 ± 0.36** | **29.08 ± 0.28** |

as the sensitive feature. It consists of 10000 samples and 200 features. Since all datasets are tabular data, for all datasets, we use different non-linear functions (e.g., sigmoid, ReLU, and Tanh) to generate two views. A succinct overview of these datasets is presented in Table 2.

**Evaluation Metrics** To evaluate the effectiveness and superiority of our DFMVC, we adopt the widely used metrics, i.e., normalized mutual information (NMI) and balance (BAL) [10, 43]. NMI is used to evaluate the clustering performance of our model, while BAL is used to assess the fairness of our model. Balance is defined as follows:

$$Balance = \min_i \frac{\min |C_i \cap s_j|}{|C_i|} \tag{13}$$

Where $C_i$ represents the $i$-th cluster, $s_j$ represents $j$-th protected subgroup. For example, for gender, there are two protected groups, namely males and females. Typically, the upper limit of balance is determined by the distribution of the sensitive feature, with a higher balance value suggesting a fairer result.

## 4.2 Experiment Setup

The experiments are executed using the following hardware configuration: Intel Core i7-9700K CPU, NVIDIA GeForce RTX 2080Ti GPU, and 64GB RAM. Additionally, all experiments make use of the Pytorch platform. In the case of DFMVC, we utilize the Adam [19] optimizer to minimize the total loss.

### 4.2.1 Comparison methods.
The proposed DFMVC is compared with the following baselines:

- K-means [31]: a method is employed to divide samples into multiple clusters, with each sample assigned to the cluster closest to it.
- DEC [46]: a deep learning approach is utilized to simultaneously learn feature representations and cluster assignments using neural networks.
- CC [23]: a clustering method based on contrastive learning, which simultaneously optimizes instance-level and cluster-level contrastive loss.
- MvDSCN [57]: a multi-view deep subspace clustering network is employed to learn a self-representation matrix from multiple views.

- DCP [28]: a multi-view clustering approach that utilizes contrastive learning to ensure consistent representations and fills in missing views using dual prediction modules.
- APADC [47]: a multi-view clustering method based on subspace learning is employed to align view distributions by minimizing disparity loss.
- MFLVC [49]: a multi-level feature learning for contrastive multi-view clustering, learns different levels of features.
- Fair-MVC [55]: a method that merges fair clustering with multi-view clustering, enhancing feature representation through the addition of contrast regularization.

### 4.2.2 Parameter Settings.
To minimize the impact of randomness, each method runs many times, and the outcomes are presented as mean values accompanied by their respective standard deviations. For the Incomplete Multi-view Clustering method, we set the missing rate to 0. In the case of our DFMVC, we adopt a two-stage training approach. We utilized the autoencoder as the pre-training model, and reconstruction loss as the loss function for parameter initialization. We assign 200 epochs for pre-training and 200 epochs for the subsequent training process. In our proposed DFMVC, a batch size of 100 is employed consistently across all datasets. The learning rate is set at 0.0001.

## 4.3 Performance Comparison(RQ1)

In this subsection, we conduct experiments to demonstrate the superiority of DFMVC with 8 baselines on four fairness datasets. In Table 3, the NMI and BAL of all the compared methods on the four datasets are presented. From these results, we have the following observations. 1) compared to classical deep multi-view clustering algorithms, our approach, DFMVC, consistently yields superior clustering results across the majority of datasets. Using the Credit Card dataset as an example, DFMVC outperforms its nearest competitors by significant margins of 8.4% and 2.48% in terms of NMI and BAL, respectively. We conjecture the reason is that the majority of methods fail to consider fairness and the learned representations are not as discriminative as desired. 2). Thanks to the consideration of fairness and the adoption of contrastive learning strategies to learn feature consistency among multiple views and obtain semantic information by contrasting clustering assignments among different views, our method can perform better.

**Table 4: Ablation study concerning the main components of the proposed DFMVC method on all datasets. The notations "(w/o) L_C" and "(w/o) L_F" denote reduced models obtained by excluding the semantic contrastive learning module and the cluster distribution-guide fair learning module, respectively.**

| Methods | $L_r$ | $L_c$ | $L_f$ | Credit Card | | Banking Market | | Law school | | Zafar | |
|---|---|---|---|---|---|---|---|---|---|---|---|
| | | | | NMI | BAL | NMI | BAL | NMI | BAL | NMI | BAL |
| (w/o) L_F | ✓ | ✓ | | 24.95 | 36.67 | 41.23 | 38.62 | 20.63 | 43.72 | 88.63 | 27.56 |
| (w/o) L_C | ✓ | | ✓ | 20.12 | 39.67 | 32.68 | 40.92 | 18.45 | 46.59 | 80.61 | 28.56 |
| DFMVC | ✓ | ✓ | ✓ | 35.13 | 39.71 | 54.62 | 42.16 | 24.24 | 45.66 | 93.93 | 29.08 |

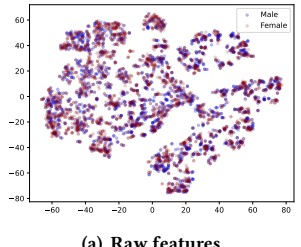 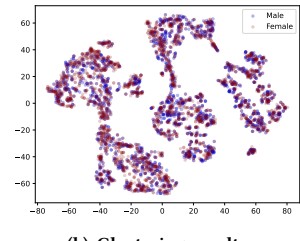 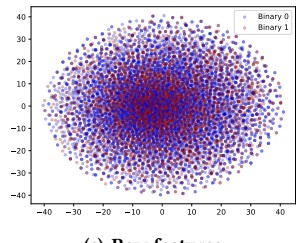 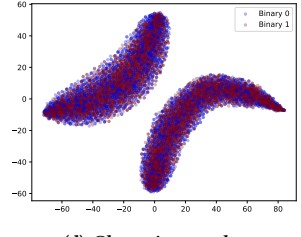

| (a) Raw features | (b) Clustering results | (c) Raw features | (d) Clustering results |

**Figure 3: Visualize sensitive features in Credit Card and Zafar dataset using t-SNE algorithm. (a-b) is on the Credit Card dataset, and (c-d) is on the Zafar dataset. On the Credit Card dataset, blue points represent male features, and red points represent female features. On the Zafar dataset, blue points represent binary 0, and red points represent binary 1. The original data is visualized on the left, and clustering results on the right.**

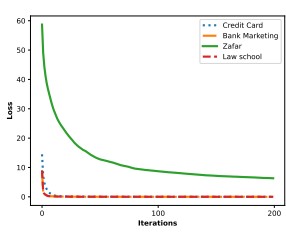 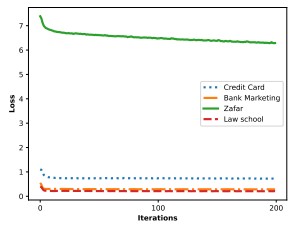

| (a) The pre-training loss curve | (b) The total loss curve |

**Figure 4: Covergence results achieved through the DFMVC method across all the datasets**

### 4.4 Ablation Studies(RQ2)

To verify the effectiveness of the semantic contrastive learning mechanism and the cluster distribution-guide fair learning module, we conduct experiments on all datasets. For simplicity, we adopt "(w/o) **L_C**" and "(w/o) **L_F**" to denote the reduced models by removing the semantic contrastive learning module and cluster distribution-guide fair learning module respectively. As shown in Table 4, the best performance can be achieved when all loss terms are considered. Take the Credit Card dataset as an example, we can observe that the clustering performance will decrease without any of our proposed modules, indicating that each module contributes to boosting the performance.

### 4.5 Visualization Analysis(RQ3)

In this subsection, we perform visualization experiments to intuitively showcase the effectiveness of DFMVC. Specifically, we employ t-SNE algorithm [37] to visualize the distribution of raw features and learned embeddings of DFMVC on the Credit Card and the Zafar dataset. Take the Zafar dataset as an example, the Zafar dataset is a widely-used synthetic dataset, and we generate a binary value as its sensitive feature. Blue dots represent binary 0, and red dots represent binary 1. From the Figure. 3(c), we can observe an even distribution of binary 0 and binary 1 in raw features. In Figure. 3(d), the binary values are evenly distributed in each cluster. This result suggests that our method demonstrates independence from sensitive attributes. That is, our method fulfills the objective of fair clustering.

### 4.6 Convergence analysis(RQ4)

In this subsection, we explore the convergence of the DFMVC method. Specifically, we explored pre-training and formal training separately. To validate the convergence of the DFMVC method, we compute the results of the loss function by Eqs. (11) and (3) during training. As shown in Figure 4, it depicts the curves of the loss function results across all the datasets. The value of the loss function defined in Eq. (3) exhibits a sharp decrease in the initial iterations, followed by a gradual decline until convergence is achieved. Similarly, we observe a comparable pattern in the variations of the loss function values defined in Eq. (11). These observations underscore the effectiveness of the convergence property inherent in the DFMVC method.

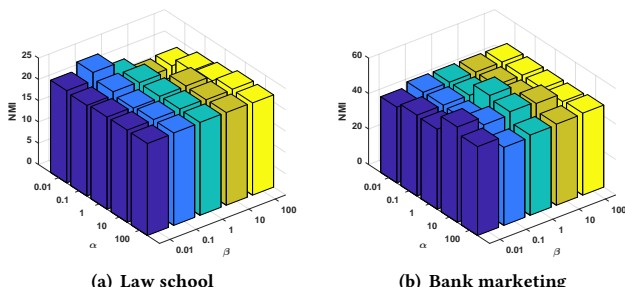

**(a) Law school**  **(b) Bank marketing**

**Figure 5: The NMI values yield by the DFMVC method with different $\alpha$ and $\beta$ combinations on the two datasets.**

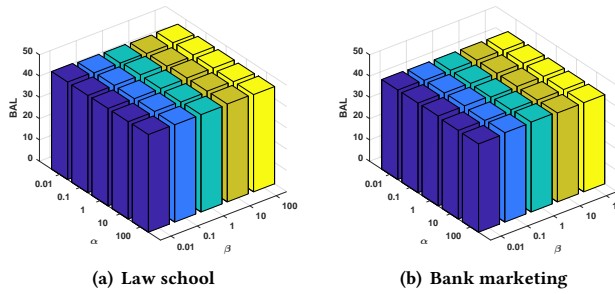

**(a) Law school**  **(b) Bank marketing**

**Figure 6: The balance value yield by the DFMVC method with different $\alpha$ and $\beta$ combinations on the two datasets.**

## 4.7 Hyper-parameter Analysis(RQ5)

We perform experiments on two representative datasets, namely the law school and bank marketing datasets, to access the impact of the $\alpha$ and $\beta$ parameters in the proposed DFMVC method. The $\alpha$ and $\beta$ are chosen from {0.01, 0.1, 1, 10, 100} for DFMVC. As can be observed in Figures 5 and 6, we can find that clustering performance and fairness achieved by the DFMVC in terms of the NMI and BAL values obtained with different combinations of $\alpha$ and $\beta$. From the figures, we can observe that different parameter combinations have a slight impact on the clustering performance but do not have much impact on the balance value. This shows that our method is not so sensitive to the selection of hyper-parameters, indicating the stability of our method.

## 5 CONCLUSION

In this paper, we propose a DFMVC method that learns more discriminative and fairer representations for MVC. Specifically, we use AutoEncoder to initialize the parameters. Then, we fuse the representations extracted from each view, and by the cluster distribution-guide fair learning module, learn fairer representations. To further use the diversity of multi-view data, we use contrastive learning to obtain feature consistency and semantic labels. We conduct extensive experiments and ablation studies on four datasets to validate the superiority of the model and the effectiveness of each component in our method.

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
