# OpenReview forum: "DFMVC: Deep Fair Multi-view Clustering"
_acmmm.org/ACMMM/2024/Conference — MM2024 Poster_

### Official Review · Reviewer_miwh · 2024-05-07

**Rating:** 6
**Confidence:** 3

**Summary:**

The paper proposes a novel fair multi-view clustering method by leveraging contrastive learning to extract consistent and discriminative features from different views, thus distinguishing multi-view data better. The work also considers the fairness issue in multi-view clustering, which is ignored by most related works and has gained widespread attention in academia, which aims to alleviate the model bias on a certain sensitive attribute set. The authors proposed to achieve the optimal fairness in multi-view clustering tasks by aligning the distribution of sensitive features with the target cluster distribution. With the carefully designed experiments on several datasets as well as the comparison with existing works, the superior clustering performance and the model fairness are well demonstrated.

**Strengths:**

The paper follows a well-structured organization. The strengths of the paper include:
(1) Fairness has received increasing attention. Despite there are some works exploring the fairness issue in multi-view learning tasks, fairness problem in multi-view clustering still remains underexplored. The work focuses on a significant research issue.
(2) The idea of the work seems technically sound. By utilizing the target cluster distribution to guide fair clustering, the unfairness problem in multi-view clustering can be effectively alleviated. The contrastive learning module helps to enforce the learned multi-view feature to be more discriminative, thus improving the clustering performance. The design is reasonable.
(3) Extensive experiments demonstrate the effectiveness of the work in achieving fairness and improving clustering performance. Comparison with existing works.

**Limitations:**

There are several limitations, summarized as follows:
(1)	In Table 1, both $H^v$ and $Z^v$ represent the embedded feature of the $v$-th view, what is the difference? Are the two symbols share the same meaning? If not, the authors should add more description to distinguish them.
(2)	The fairness of the proposed method is evaluated via experiments on several fairness datasets. However, the method in this work to achieve fairness, i.e., by aligning the distribution of sensitive attributes with the cluster distribution, should be further discussed by providing more justifications.
(3)	The paper should be proofread to improve the usage of English. Please go through this paper carefully, and correct all the typos.

**Suitability:**

3

---

### Official Review · Reviewer_Szfn · 2024-05-15

**Rating:** 5
**Confidence:** 3

**Summary:**

This paper presents a novel deep fair multi-view clustering method, which attempts to learn a consistent and discriminative representation for clustering guided by a fairness constraint constructed from clustering distribution. It aims to address the unfairness issue existing in current multi-view clustering methods, which is a significant research topic. Its main theoretical contributions include: (1) It incorporates a contrastive constraint to learn a consistent representations of multi-view data, which can better obtain the discriminative information to improve clustering performance; (2) By aligning the distribution of sensitive features with the target cluster distribution, it well ensures fairness and meanwhile maintains the satisfactory clustering results. Besides, extensive analyses and experiments are performed, whose results well demonstrate the method’s superiority.

**Strengths:**

1.	The research topic is significant, since fairness attracts more and more attention, while this issue still lacks exploration in multi-view clustering. The paper is relatively well-written with reasonable organization and clear motivations. The literature review covers most related works in terms of multi-view clustering and fairness and provides a deep analysis of them.
2.	The work is technically sound, which employs contrastive learning to enhance the discriminativeness of the consistent feature extracted from different views, and by introducing a fairness constraint constructed from the sensitive attribute distribution in different clusters, it well ensures the fairness. The method design is reasonable. Sufficient experiments are conducted, and by comparing the proposed method with existing multi-view clustering algorithms, its effectiveness and superiority in both multi-view clustering performance and fairness is well demonstrated.

**Limitations:**

1.	Although this paper is overall well-written, there still exist some typos and grammar mistakes. Some details still need to be amended. For example, the notation $Z^G$ in Table 1 is not clearly defined or introduced, and maybe it should be a mistake and should be revised as $Z$.
2.	Figure 2 should be further improved. The symbol $D_2$ and $E_2$ in it seem to be the decoder and encoder for the $n$th view, and thus they should be $D_n$ and $E_n$.
3.	In Figure 3, the authors use different colors to represent different sensitive attributes, which can effectively demonstrate the fairness of the proposed method. However, the selected two colors should be changed because these two colors make readers not able to well distinguish them.

**Suitability:**

3

---

### Official Review · Reviewer_seXE · 2024-05-20

**Rating:** 5
**Confidence:** 3

**Summary:**

The paper is concentrated on the deep fair multi-view clustering. The authors first introduce the unfairness problem confronted by existing multi-view clustering, as illustrated by an example in Figure 1. Then, a distribution alignment based solution is developed, which leverages target cluster distribution to guide fair clustering. To simultaneously achieve high clustering performance, a contrastive learning module is introduced to obtain the consistent and discriminative representation. The improved discriminativeness helps to better sort samples into different clusters. In this way, the method achieves both fairness and clustering results. To support the conclusion of the paper, the authors also conduct a number of experiments, whose results indicate the superiority of the proposed method. In summary, the paper is technically sound, and the conclusion are well supported by the theoretical analysis and experimental results.

**Strengths:**

(1) An innovative deep multi-view clustering model is developed. It is interesting and novel to leverage target cluster distribution to guide fair clustering in deep multi-view clustering, which is a practical scenario in real-world applications, for the multi-view clustering task. Figure 1 also clearly shows the research motivations.
(2) The technique is sound and useful for the multi-view clustering tasks. First, it employs contrastive learning to enhance the discriminative features extracted. Second, it leverages target cluster distribution to guide fair clustering, leading to that the clustering results are independent of sensitive attributes.
(3) Extensive comparison experiments on four real world datasets with sensitive attributes have been conducted to prove the effectiveness of the proposed model. Compared to the state-of-the-art methods, the proposed deep fair multi-view clustering method achieves promising clustering performance as well as fairness.

**Limitations:**

(1)	For the writings, there are several typos to be corrected. Some symbols are not well defined. The authors should carefully check the entire paper and revise it carefully.
(2)	For the experiments, the authors describe each dataset from the perspective of sensitive attributes. However, the different views of these datasets are less introduced. The authors should explain each view of the four datasets adopted by the experiments.
(3)	Some details require more introduction. For example, during the adaptive fusion, how to initialize $A$ is not well illustrated. For another, how to obtain the final clustering result?

**Suitability:**

3

---

### Official Review · Reviewer_SHer · 2024-05-24

**Rating:** 5
**Confidence:** 3

**Summary:**

The paper proposes a new deep fair multi-view clustering method called DFMVC. Mining inter-view consistency information and preserving discriminative information of each view based on contrastive learning and reconstruction loss. Divide subgroups by sensitive attributes and generate target soft distributions within the subgroups to guide clustering optimization to achieve fair clustering.

**Strengths:**

1.	The paper is clearly structured and easy to read.
2.	The idea is very simple and novel.
3.	Extensive experiments on multiple datasets confirm the effectiveness of the proposed method.

**Limitations:**

1.	The description in Figure 1 is not very clear. What does a sample without sensitive attributes refer to?
2.	There are some errors. For example, in the last paragraph of the introduction section, the framework diagram should be Figure 2 instead of Figure 1.
3.	hj in Equation 4 is the j-th column of the H matrix, and the H matrix is N*K, which indicates that Equation 4 uses all samples at the same time to calculate similarity. Since the training process is based on min-batch, please explain how this is implemented. If you are calculating the similarity within min-batch, since Equation 4 is batch size sensitive, additional batch size ablation experiments are required.
4.	Why does the denominator in Equation 5 have -e^1/τ, and is there a lack of positive terms in the denominator?
5.	αv in Equation 7 is the same for all samples of view v, which assumes that all samples in the same view have the same importance.
6.	The subscripts of Σ in Equation 10 are confusing.

**Suitability:**

3

---

### Official Review · Reviewer_H6iw · 2024-05-26

**Rating:** 5
**Confidence:** 3

**Summary:**

The main idea of this paper is to explore the fairness in deep multi-view clustering. The authors proposed Deep Fair Multi-View Clustering (DFMVC) method that learns a consistent and discriminative representation instructed by a fairness constraint constructed from the distribution of clusters. It incorporates contrastive constraints on semantic features from different views to obtain consistent and discriminative representations for each view. The proposed method aligns the distribution of sensitive attributes with the target cluster distribution to achieve optimal fairness in clustering results. Experimental results on the multi-view datasets that contains sensitive attributes demonstrate the validity of the proposed deep fair multi-view clustering method and its superior advantages over several state-of-the-art algorithms.

**Strengths:**

(1) The idea of this paper is innovative, and the main advantages of the proposed method are designing a novel deep fair multi-view clustering method. The proposed method explores the unfairness problem of multi-view clustering. The cluster distribution is utilized to guide the distribution of sensitive attributes, which alleviates the problem that samples with certain sensitive attributes are incorrectly clustered.
(2) The effectiveness of the method is somehow obvious. Contrastive learning is a popular method to learn better features, with which the clustering result can be improved. For fairness, aligning distribution via KL divergence is consistent with the conceptual definition of fairness in machine learning. Thus, the work’s effectiveness holds in theory.
(3) Extensive experiments on four datasets are conducted, and the performance in terms of fairness and clustering is evaluated with the established metrics.

**Limitations:**

(1) The literature review seems not enough, since there are many works on deep multi-view clustering. More should be involved and analyzed, or the motivations of the work are not convincing.
(2) The authors only compare the proposed method with one fair multi-view clustering method (Fair-MVC). Although there are few fair multi-view clustering methods, fair clustering methods in single views have been widely studied, resulting in many related works proposed. Therefore, to better demonstrate the superiority of the work, more single-view fair methods are recommended to be compared.
(3) It is noted that the paper needs careful editing by someone with expertise in technical English editing paying particular attention to English grammar, spelling, and sentence structure so that the goals and results of the study are clear to the readers.

**Suitability:**

3

---

### Meta-Review · Area_Chair_yJnL · 2024-06-28

**Recommendation:** Accept (Poster)
**Confidence:** 5

**Metareview:**

The existing fair multi-view clustering methods are built on the uniform distribution of attributes. In contrast, this paper constructs a fairness constraint by using the distribution of clusters, which can preserve the consistent and discriminative representation across different views. They used the contrastive loss on semantic features and alignment loss on the distribution of sensitive attributes. Many experiments show the effectiveness of the proposed method against different MVC algorithms.

We have five reviewers who carefully checked this paper. They raise many concerns about the technical and experimental details as well as the fairness description. After rebuttal, all the reviewers are satisfied with the current form of this paper. All of them voted to accept this paper. Considering the good presentation, structure, and experimental results, this paper deserves to be accepted to ACM MM 2024.